# Feeding behavior and activity of *Phlebotomus pedifer* and potential reservoir hosts of *Leishmania aethiopica* in southwestern Ethiopia

**Myrthe Pareyn** [1] ⊕ *, **Abena Kochora** [2] ⊕, **Luca Van Rooy** [1], **Nigatu Eligo** [2], **Bram Vanden Broecke** [1], **Nigatu Girma** [2], **Behailu Merdekios** [3], **Teklu Wegayehu** [2], **Louis Maes** [4], **Guy Caljon** [4], **Bernt Lindtjørn** [5], **Herwig Leirs** [1], **Fekadu Massebo** [2] *

**1** Evolutionary Ecology Group, University of Antwerp, Antwerp, Belgium, **2** Biology Department, Arba Minch University, Arba Minch, Ethiopia, **3** Public Health Department, Arba Minch University, Arba Minch, Ethiopia, **4** Laboratory of Microbiology, Parasitology and Hygiene, University of Antwerp, Antwerp, Belgium, **5** Centre for International Health, University of Bergen, Bergen, Norway

⊕ These authors contributed equally to this work.
* myrthe.pareyn@uantwerpen.be (MP); fekadu.massebo@amu.edu.et (FM)

**Data Availability Statement:** All relevant data are within the manuscript and its Supporting Information files.

## Abstract

### Background

Cutaneous leishmaniasis (CL) is a major public health concern in Ethiopia. However, knowledge about the complex zoonotic transmission cycle is limited, hampering implementation of control strategies. We explored the feeding behavior and activity of the vector (*Phlebotomus pedifer*) and studied the role of livestock in CL transmission in southwestern Ethiopia.

### Methods

Blood meal origins of engorged sand flies were determined by sequencing host DNA. A host choice experiment was performed to assess the feeding preference of *P. pedifer* when humans and hyraxes are equally accessible. Ear and nose biopsies from livestock were screened for the presence of *Leishmania* parasites. Sand flies were captured indoor and outdoor with human landing catches and CDC light traps to determine at which time and where *P. pedifer* is mostly active.

### Principal findings

A total of 180 *P. pedifer* sand flies were found to bite hosts of 12 genera. Humans were the predominant blood meal source indoors (65.9%, p < 0.001), while no significant differences were determined outdoors and in caves. In caves, hyraxes were represented in blood meals equally as humans (45.5% and 42.4%, respectively), but the host choice experiment revealed that sand flies have a significant preference for feeding on hyraxes (p = 0.009). Only a single goat nose biopsy from 412 animal samples was found with *Leishmania* RNA. We found that *P. pedifer* is predominantly endophagic (p = 0.003), but occurs both indoors

**Funding:** This work was supported by the University of Antwerp and the Norwegian Programme for Capacity Development in Higher Education and Research for Development (NORHED). MP is a fellow of the Flemish Interuniversity Council (VLIR-UOS, NDOC2016PR003). The funders had no role in the study design, data collection and analysis, decision to publish, or preparation of the manuscript.

**Competing interests:** The authors have declared that no competing interests exist.

and outdoors. A substantial number of sand flies was active in the early evening, which increased over time reaching its maximum around midnight.

## Conclusion

In contrast to earlier suggestions of exclusive zoonotic *Leishmania* transmission, we propose that there is also human-to-human transmission of CL in southwestern Ethiopia. Livestock does not play a role in CL transmission and combined indoor and outdoor vector control measures at night are required for efficient vector control.

## Author summary

Cutaneous leishmaniasis is a major public health problem in Ethiopia. It is mainly caused by *Leishmania aethiopica* protozoa that are transmitted when female sand flies take a blood meal. Hyraxes are assigned as the reservoirs of the infection, because many were found infected with *Leishmania*. There is very limited knowledge about the behavior of sand flies and other potential reservoir hosts of the infection. However, this information is a prerequisite for disease control, which is currently hampered. In this study, we found that humans are an important source of infection and that the role of hyraxes in disease transmission needs further investigation to decide whether they should be included in control programs. Livestock appears not play a role in transmission, even though sand flies like to feed on them. We also show that sand flies are active indoors and outdoors, but have a preference for feeding inside human dwellings and that they are mostly active around midnight. Overall, we conclude that disease prevention and control should emphasize on human protection by applying vector control indoors, at night.

## Introduction

Cutaneous leishmaniasis (CL) is a vector born disease, caused by *Leishmania* protozoa and transmitted by female phlebotomine sand flies. It is characterized by nodules or ulcerative skin lesions on people's faces and extremities, which result in disfiguring scars after healing [1,2].

CL is a major public health concern in Ethiopia, affecting approximately 20,000 to 50,000 people annually [3], in which *Leishmania aethiopica* is responsible for the majority of the infections [4–6]. Ochollo, our study site, is a village in the mid-highlands of southwestern Ethiopia, where CL is endemic and is mainly affecting young children [7,8]. A recent study identified 4% of the primary school children with active lesions, 1.5% with lesions and scars and 59.8% with scars [8]. Adults are very seldom found with active lesions, because they already recovered from a childhood *Leishmania* infection, thereby becoming resistant to the development of clinical infection [9]. There are currently no control programs for CL in southern Ethiopia, mainly because of the complexity of the zoonotic transmission cycle and the limited understanding of the vector's behavior.

Previous researchers described *Phlebotomus pedifer* as the only vector in Ochollo [10–13], showing a 3.5% infection rate [13]. A study in Kenya found a high susceptibility of *P. pedifer* to *L. aethiopica* when feeding on active human CL lesions, implying that it could be an efficient vector [14]. The species has been found indoors, around household compounds, in tree holes, rocky areas and inside caves [12,13]. A study in Ochollo in 1973 showed that 11 *P. pedifer* sand flies from indoors and five from caves were solely feeding on humans and hyraxes respectively

[12]. However, until now relatively little is known about its biting behavior. Sand flies are generally known to be active between dusk and dawn and females feed on a wide variety of vertebrate hosts. However, the peak activity and host preference differs among sand fly species, so species specific entomological data are crucial to obtain a clear image of the transmission cycle [15–17].

Besides the vector, the reservoirs of the infection should be well documented. Hyraxes (*Heterohyrax bucei* and *Procavia capensis*) have been described as the reservoir of the zoonotic transmission of CL in Ethiopia [4–6,13]. *H. brucei* is abundant in Ochollo and a large proportion has been found infected with *L. aethiopica*. They live near human settlements, in caves and rock crevices, where sand flies and other potential hosts are abundant [12,13]. Rodents were found most probably not to play a role in transmission in Ochollo [13], but other animals have so far not been investigated yet as carriers of *L. aethiopica*. Given that bovines are commonly bitten by the main CL vector in Ethiopia, *P. longipes*, their role in transmission needs further investigation [12,18,19].

Successful disease control requires profound understanding of the transmission cycle. Knowledge about the blood meal preference of sand flies is crucial to demonstrate which vertebrates might contribute to disease transmission and should be included in control programs. Moreover, information on where and at what time sand flies are biting is a prerequisite to decide which vector control methods should be applied.

In this study, we aimed to gather knowledge on *(i)* the blood meal sources of *P. pedifer* in different habitats and its feeding preference when hosts are equally available, *(ii)* the role of domestic animals in CL transmission, and *(iii)* the indoor and outdoor activity pattern of *P. pedifer*. This information will shed light on the natural transmission cycle of CL in southwestern Ethiopia and help in instructing control efforts in the area.

## Methods

### Ethics statement

This study was reviewed, approved and monitored by the Institutional Ethics Review Board (IRB) of Arba Minch University (cmhs/1203482/111 and cmhs/120017/111). Healthy adults (> 18 years) with obvious scar formation, who have been living in Ochollo their whole life, were selected as subjects for the human landing collections and host choice experiment. Written informed consent was obtained from all human volunteers who participated. All animal handlings were carried out according to the 2016 Guidelines of the American Society of Mammologists for use of mammals in research and education and in agreement with the appropriate institutional authorities.

### Study area

Ochollo is located in southwestern Ethiopia (6˚11’N, 37˚ 41’ E), about 20 km North of Arba Minch (Fig 1). It is a rocky area with steep slopes and basalt cliffs with caves, situated at an altitude ranging between 1600 m and 2200 m. The area has a modest climate with an average yearly temperature around 20˚C and a high humidity from May until October [13]. The village covers approximately 1100 hectares and is divided into eight sub-villages. Ochollo is densely inhabited by approximately 5000 people, which are mainly clustered on the tops of hills and steep slopes. People ranch cattle and goats, and some households have dogs. Hyraxes are abundant and live in caves and rocky areas near human residences, while rodents mainly occupy stone fences and human and animal dwellings. Houses are mainly made of mud, wood and grass, leaving many openings for sand fly entry and resting places.

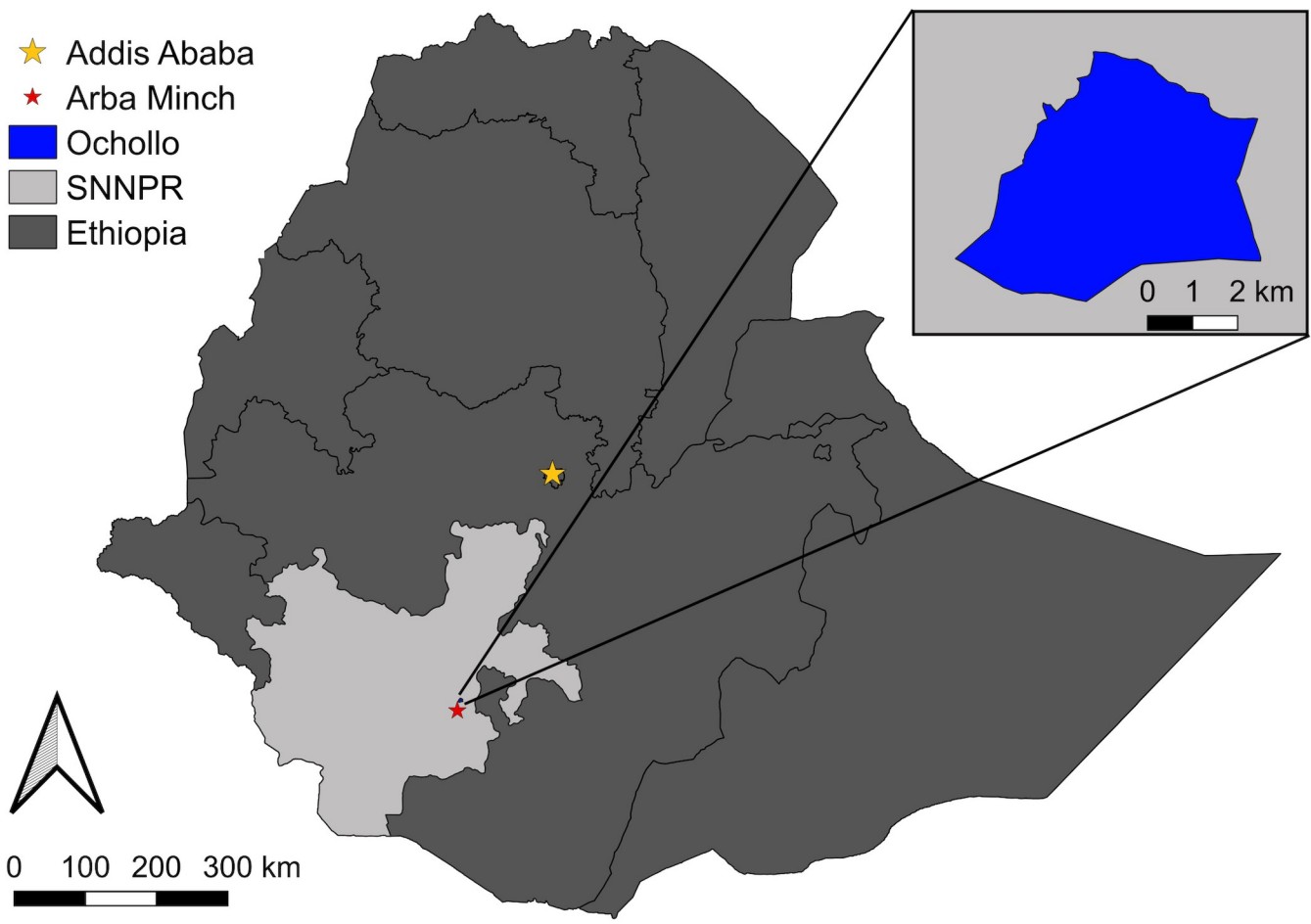

**Fig 1. Map of the location of the study site, Ochollo, in Ethiopia [20,21].** SNNPR: Southern Nations, Nationalities and People's Region.

## Host identification

**Sand fly collection.**   Sand flies were collected indoors and outdoors from 72 households (nine households in each of the eight sub-villages) between February and May 2018. Additionally, ten caves were selected in the village for monthly sand fly collections from March to June 2018. Caves were located within the village, in a range of maximum 300 meters from human dwellings. Trapping was performed once per month at each sampling site with a particular entomological approach. Indoors and in caves, one CDC miniature light trap (John W. Hock Company, Florida, USA) and five sticky traps (ST, A4 format white papers attached to card board, covered with plastic impregnated with sesame oil on both sides) were placed at the bed end and wall cracks indoors, and inside caves. Collection with the two methods was performed on separate days. Only ST were utilized outdoors (N = 5 per collection site), which were placed on wall cracks of the houses and surrounding potential sand fly breeding or resting sites. Traps were set at 18h and collected again the next morning at sunrise. Blood fed female sand flies were sorted out, and the thorax and abdomen were dissected and stored in 97% ethanol at -20˚C until further analysis. No distinction was made among different stages of blood digestion in sand flies.

**Blood meal analysis.**   DNA isolation from the blood fed specimens was performed with a NucleoSpin Tissue kit (Macherey Nagel, Düren, Germany) according to the manufacturer's

instructions. Finally, the DNA was eluted in 50µl nuclease free water. Unfed sand flies and sand flies fed on laboratory mice (*Lutzomyia longipalpis*, acquired from the Laboratory of Microbiology, Parasitology and Hygiene, University of Antwerp, Belgium) were respectively used as negative and positive extraction controls. DNA extracts were subjected to a PCR targeting a fragment of the Cytochrome B gene (*Cyt B*, 359 bp) as described by Steuber *et al.* (2005) and Carvalho *et al.* (2017) [22,23]. In short, the 15µl reaction mixture consisted of 1X Green GoTaq Flexi buffer (Promega, Leiden, Netherlands), 1.5mM MgCl$_2$ (Promega, Leiden, Netherlands), 0.5µM of both primers Cyt1 (5'-CCA TTC AAC ATC TCA GCA TGA TGA AA-3') and Cyt2 (5'-GCC CCT CAG AAT GAT ATT TGT CCT CA-3')(Life Technologies, Merelbeke, Belgium), 0.2 mM dNTPs (GE Healthcare Lifescience, Diegem, Belgium), 1U GoTaq G2 Flexi DNA Polymerase (Promega, Leiden, Netherlands) and 1.5µl DNA template. Amplification was carried out with an initial activation step of two minutes at 95˚C, followed by 40 cycles of 30 seconds at 94˚C, 30 seconds at 52˚C and one minute at 72˚C, and a final extension step of five minutes at 72˚C. The PCR was performed on Biometra T professional gradient Thermocycler (Biometra, Westburg, Netherlands). Positive and negative PCR controls and the above-mentioned extraction controls were included in each of the PCR reactions. PCR results were visualized on a 1.5% gel. After the PCR analyses were carried out at Arba Minch University in Ethiopia, the amplicons were sent to the Vlaams Instituut voor Biotechnologie (VIB) at the University of Antwerp in Belgium for sequencing. The obtained *Cyt B* sequences were aligned in GenBank using BLAST to determine the host species that served as a blood source. Results were only included when both query coverage and identity exceeded 95%.

**Sand fly species identification.**    If the blood meal of a specimen was successfully identified, the sand fly species was determined with a PCR targeting a 700 bp fragment of the *cytochrome c oxidase subunit I* (*COI*) gene, as described by Kumar *et al.* (2012) and Pareyn *et al.* (2019) [13,24].

*Leishmania* **screening.**    Sand flies of which the blood meal was successfully determined were subjected to a real-time PCR assay targeting kinetoplast DNA (kDNA) for *Leishmania* detection. Furthermore, the *Leishmania* species of the kDNA positive specimens was determined with a PCR targeting a 350 bp fragment of the internal transcribed spacer 1 (ITS-1) gene followed by sequencing. Both assays were carried out as described in our previous study [13].

**Livestock sample collection.**    Ear and nose biopsy samples from livestock (bovines and goats) were collected between January and April 2019. Samples originated either from animals that were slaughtered for human consumption or live animals. For the latter collection method, Xylocaine 2% gel (Astra Zeneca, Dilbeek, Belgium) was applied on the nose and ear for local anesthesia. Samples were collected using a 3 mm Biopsy puncher (Henry Schein, Vilvoorde, Belgium) and stored in 97% ethanol at -20˚C until further analysis. To stop the bleeding, the incised skin wound was ligated with skin glue. Between ear and nose biopsy collections of each animal, the puncher was cleansed with 1% bleach and rinsed in distilled water, and a new puncher was used for each animal.

*Leishmania* **detection in livestock.**    Nose and ear biopsies were screened for the presence of *Leishmania* nucleic acids at Arba Minch University (Ethiopia). Samples of each animal were subjected to a reverse transcriptase real-time PCR (RT-qPCR) targeting Spliced-Leader (SL-) RNA [25]. Additionally, a selection of the samples (216/412) was also screened for the presence of kDNA to confirm the results of the first assay [13]. Nose and ear biopsy samples of each animal were pooled before extraction. Both RNA and DNA were isolated from the selection of samples that were tested by the two assays using the NucleoSpin RNA kit (Macherey-Nagel, Düren, Germany) and NucleoSpin RNA/DNA buffer set (Macherey Nagel, Düren, Germany)

following the manufacturer's instructions. For the remaining samples, only RNA isolation was carried out using solely the first method. A naive and *L. major* infected mouse ear from an experimental infection (Laboratory of Microbiology, Parasitology and Hygiene, University of Antwerp, Belgium; ethical approval UA ECD 2017–80) were respectively used as negative and positive extraction controls. The SL-RNA and kDNA qPCR assays were carried out as described previously by Eberhardt *et al.* and Pareyn *et al.* respectively [13,25]. Extracts were 1:10 diluted before addition to the reaction mixture to avoid inhibition. Each PCR run included a no-template (negative) control and an infected (positive) control (a *L. aethiopica* infected hyrax nose biopsy DNA/RNA extract). For positive animals, ear and nose tissue samples were subjected to a separate extraction, followed by a PCR to determine the parasite RNA/ DNA presence in the different tissues. Finally, the SL-RNA and kDNA positive samples were subjected to a PCR targeting ITS-1 for *Leishmania* species identification.

## Sand fly activity pattern

**Human landing catches.**    To assess the indoor and outdoor human biting rhythms of *P. pedifer*, human landing catches (HLC) were conducted in and around four different household compounds between March and August 2018. HLC were done once per month at each sampling site, indoors and outdoors on the same day. Four collectors were used for each collection night, each working for six hours. In the first part of the night (between 18h and 24h), one collector performed the HLC indoors and one outdoors. In the second part of the night (24h until 6h), two other collectors carried out the same activities. The person sitting outdoors was positioned at least 10 m from the house. The collectors sat on chairs with only their legs exposed and sand flies that landed on their legs were collected using a mouth aspirator. For each iteration of the experiment, there was an exchange in pre- and post-midnight and indoor-outdoor shifts to compensate for individual differences in attractiveness and collection skills. The collected specimens were cleared in Nesbitt's solution, washed with 70% ethanol and mounted in Hoyer's medium. Sand flies were determined up to species level with morphological identification keys [11,26].

**CDC light trap captures.**    Sand fly activity was also studied with CDC miniature light traps (John W. Hock Company, Florida, USA) from January to March 2019. Sand flies were captured during eight trapping sessions in a cave, which was previously determined as a hotspot of *P. pedifer* [13]. Traps were placed between 18h and 18h30 and collections were performed with an interval of 75 minutes until about 1h, eventually resulting in five collections per night. Sand flies were collected with a mouth aspirator and the number of male and female sand flies was recorded to establish the hourly activity of the sand flies. Captured sand flies were later used for the host choice experiment.

**Host choice experiment.**    A host choice experiment was carried out using the sand flies that were captured with CDC light traps for the sand fly activity assessment [27]. Additional sand flies were trapped in other surrounding caves in Ochollo with CDC miniature light traps (John W. Hock Company, Florida, USA) in order to increase the sample size. The experiment was performed at 2h, immediately after the hourly CDC light trap captures.

The experimental set-up consisted of three connected cages (Fig 2) [27]. Female non-fed sand flies were placed in the middle cage. In each of the lateral cages, a particular host served as a blood meal source: a human volunteer's hand and forearm and a hyrax. Hyraxes were trapped by local people using traditional trapping methods. They were sedated during the experiment with ketamine (10mg/kg intramuscularly; Verve Human Care Laboratories, Uttarakhand, India) and placed inside the cage exposing the nose, ears and forepaws to the sand flies. Their eyes were protected from sand fly bites with a napkin. Human volunteers were

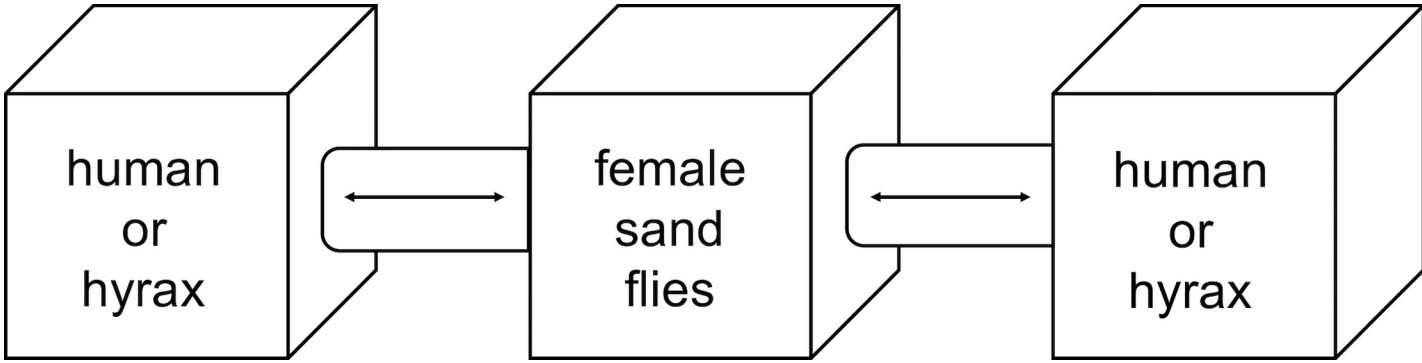

**Fig 2. Host choice experiment set-up.** Female non-fed sand flies captured from caves were transferred to the middle cage, where they were left for 30 minutes to adapt. Then, the connecting tubes to the lateral cages, where a hyrax and human hand were exposed, were opened to allow sand flies to obtain their preferred blood meal during four hours. The hosts themselves and their places were changed for each iteration of the experiment.

lying down in a horizontal position with their head nearby the cage for $CO_2$ attractiveness towards the sand flies. The distance between the two hosts was approximately three meters and the experimental set-up was covered by a plastic canvas to avoid interference of wind and other potential feeding sources in the surroundings.

After 30 minutes adaptation in the middle cage, the connecting tubes to the two lateral cages were opened for four hours to allow sand flies to bite their preferred host. Blood fed sand flies were collected using a mouth aspirator and stored in 97% ethanol at -20˚C until further analysis. The experiment was conducted eight times and for each iteration, the position of the hosts was changed and new subjects were used. Hyraxes were released at their trapping site after the experiment. The blood meal sources and sand fly species were determined by sequencing a fragment of the *Cyt B* and *COI* gene respectively, according to the methods described above (blood meal analysis and sand fly identification).

## Data analysis

All statistical analyses were carried out in R version 3.5.0, using packages "lme4" and "lmerTest" [28,29]. P-values < 0.05 were assumed statistically significant.

**Sand fly blood meal sources.**   To assess which host group served as an important blood meal source for sand flies, a generalized linear mixed model (GLMM) with Poisson error distribution was used. The number of sand flies that fed on a particular host, within a specific habitat, during a certain month was included as the response variable. The habitat where sand flies were captured (indoors, outdoors, cave), the host group they acquired their blood meal from and the interaction between habitat and host type were included as fixed effects. In order to correct for monthly variation in sand fly presence, we incorporated the collection as a random effect in the model. A post hoc test, specified as Tukey test, was applied to compare the hosts groups with each other [30].

After the previous general model, GLMMs were made similarly for each habitat separately to determine the important blood meal sources in each habitat. The model was constructed as described above, but only the host group was included as a fixed variable.

**Sand fly activity.**   The sand fly human biting rhythms indoors and outdoors were measured with HLC. We had to transform our data to a binomial distribution (0 = no sand flies were caught within an hourly time interval, 1 = one or more sand flies were captured within a time interval), because the hourly counts were low. A GLMM was used with the HLC at each time interval as the dependent variable with a binomial error distribution. The location where

sand flies were trapped (indoor/outdoor) was included as a fixed effect in order to assess if sand flies bite significantly more indoors compared to outdoors. Time interval was included as a fixed effect to compare at which moment sand flies were mostly active. We used the time interval as a categorical variable instead of continuous, because our preliminary analyses showed that there was a non-linear correlation. The trapping month and sampling site were incorporated as random effects in the model.

Sand fly activity was based on the number of captured sand flies with a CDC light trap. A GLMM with Poisson error distribution was used to estimate the sand fly activity. Due to technical difficulties, we were not able to include the fifth trapping night in the final dataset. The number of male and female sand flies at a certain time interval was used as the dependent variable. Sex was implemented as a fixed effect in the model to assess whether there were more males or females. Time interval was included as a categorical fixed effect to determine during which period most individuals were present. The experiment day was incorporated as a random effect to correct for potential differences between the sampling days.

**Host choice experiment.** A GLMM with binomial distribution was used to determine the preferred blood meal source of sand flies when both humans and hyraxes are available. The proportion of sand flies that either fed on a hyrax or human during a single experiment was used as the response variable. The proportions were weighed by the total amount of sand flies that took a blood meal within a single experiment, since this varied between the different experiments. The type of host and its position in the experiment were included as fixed effects to establish which host was preferred for a blood meal while correcting for potential personal and environmental bias. The experiment day was included as a random effect to correct for variation between days.

## Results

### Sand fly blood meal sources

A total of 11,488 sand flies were collected, of which 368 were blood fed female sand flies, which underwent procedures for blood meal origin identification. 92 (25.0%) of these samples were excluded from the analysis, as negative extraction controls tested positive, indicating contamination during the DNA isolation procedure. The *Cyt B* gene could not be amplified for 11 samples (3.0%) and the sequence identity of 83 samples (22.6%) could not be determined using the previously set cut-off requirements for the BLAST analysis. The overall analysis resulted in successful blood meal identification for 182 (49.5%) specimens. All of these specimens, except for two, turned out to be *P. pedifer*. The other two matched with several sand fly species of the subgenus *Laroussius* in GenBank with low query coverage and identity, but could not be identified up to species level. One sand fly acquired its blood meal from a human and the other one from a bush hyrax.

A total of 180 *P. pedifer* sand flies fed on 12 different hosts, presented in Table 1. Overall, humans were the most important blood meal source (p < 0.001), accounting for 59.4% of the identified origins, followed by bovines (13.9%), bush hyraxes (10.6%), goats (7.2%) and rodents (5.0%). Residual blood meals were acquired from a wide variety of vertebrates, together covering 4.0% of the determined sources. From the sand flies that fed on humans, five out of 107 (two collected from caves and three from indoors) were positive for *Leishmania* kDNA, which were all *L. aethiopica* infections.

Indoors, 129 blood fed sand flies were collected (Fig 3A). Significantly more sand flies (65.9%, p < 0.001) had fed on humans (S1 Table) compared to 23.3% that fed on livestock (16.3% on bovines and 7.0% on goats) and 5.4% on rodents. Three of the sand flies that were captured indoors had acquired their blood meal from hyraxes.

**Table 1. Blood meal analysis of _Phlebotomus pedifer_ in Ochollo between February and May 2018.** Scientific and common name of blood meal sources grouped into categories for further analysis, and the number and percentage of sand flies that fed on each host. Out of 180 sand flies, 129 were collected indoors, 18 outdoors and 33 inside caves.

| Host category | Scientific name | Common name | Number | Percentage |
|---|---|---|---|---|
| **Human** | _Homo sapiens_ | Human | 107 | 59.4% |
| **Livestock** | _Bos taurus_ | Bovine | 25 | 13.9% |
| | _Capra hircus_ | Goat | 13 | 7.2% |
| **Hyrax** | _Heterohyrax brucei_ | Bush hyrax | 19 | 10.6% |
| **Rodent** | _Acomys spp._ | Spiny mouse | 6 | 3.3% |
| | _Grammomys sp._ | Thicket rat | 2 | 1.1% |
| | _Arvicanthis sp._ | Grass rat | 1 | 0.6% |
| **Other** | _Myonycteris angolensis_ | Bat | 2 | 1.1% |
| | _Canis lupus familiaris_ | Dog | 2 | 1.1% |
| | _Gallus gallus_ | Chicken | 1 | 0.6% |
| | _Felis catus_ | Cat | 1 | 0.6% |
| | _Tragelaphus sp._ | Bushbuck | 1 | 0.6% |
| **Total identified** | | | 180 | |

No significant difference in blood meal sources could be determined from the 18 sand flies that were captured outdoors (Fig 3B, S1 Table), but the most important origins were again humans (38.9%) and livestock (33.3%).

In caves, 33 sand flies were blood fed, but no significant differences in blood meal sources were determined (Fig 3C, S1 Table). Hyraxes were represented in blood meals equally as humans (45.5% and 42.4% respectively).

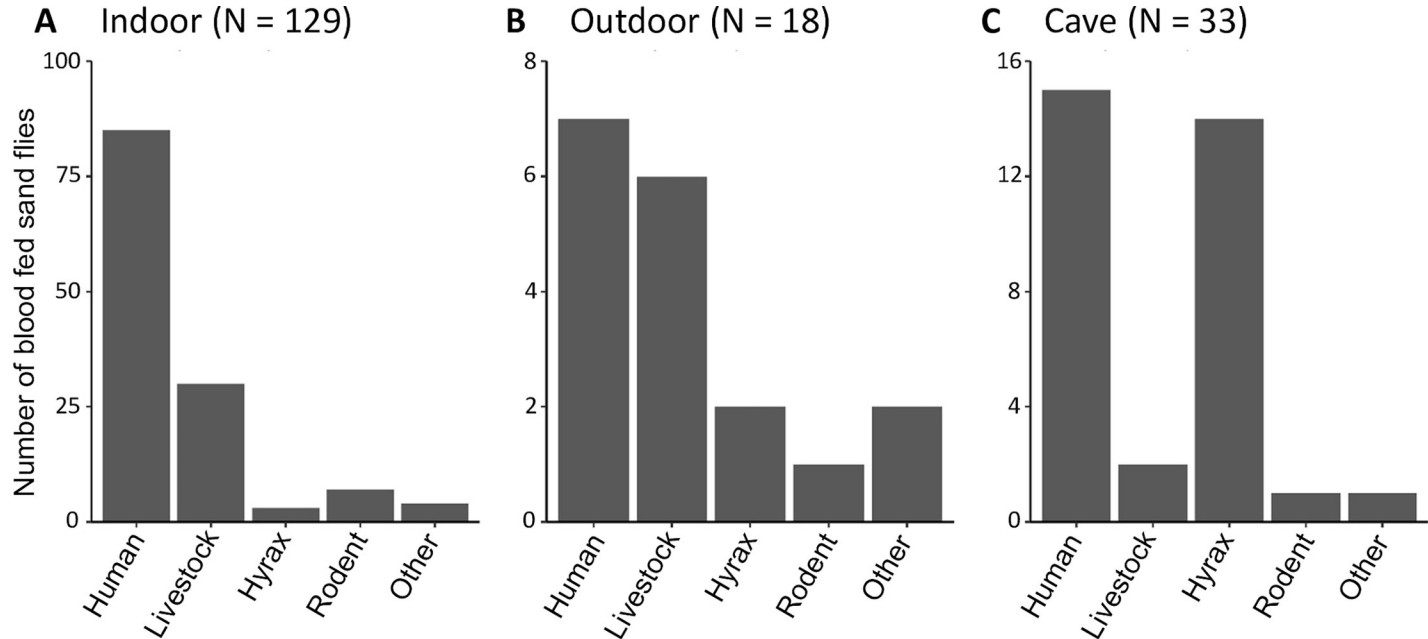

**Fig 3. Blood meal sources of _Phlebotomus pedifer_ captured in different habitats in Ochollo 2018.** Blood meal sources of sand flies captured (A) indoors (B) outdoors and (C) in caves. The category livestock includes bovines and goats and the category rodents consists of _Acomys spp._, _Grammomys sp._ and _Arvicanthis sp._ The 'other' host group includes all other vertebrates that sand flies fed on (Table 1).

### *Leishmania* in livestock

A total of 412 ear and nose samples, of which 209 from bovines and 203 from goats, were collected. Of the 412 samples, 17 were collected from slaughtered bovines and 395 from live animals. The selection of the samples that were subjected to both kDNA and SL-RNA assays were all negative. The pooled sample of one live goat was positive for SL-RNA. After separate tissue extractions of this goat, the nose sample appeared positive for kDNA and SL-RNA, with a Ct value of approximately 28 in both assays. The ITS-1 gene could not be amplified for *Leishmania* species identification, presumably due to a low parasitemia.

### Sand fly activity

**Human landing catches.** A total of 161 sand flies were captured with HLC, of which 93% were identified as *P. pedifer*, while the remaining 7% belonged to the subgenus *Sergentomyia*, which were removed from further analysis.

*P. pedifer* was found to bite humans both indoors and outdoors, but the overall the probability of indoor biting was significantly higher (p = 0.003, Fig 4A).

Sand fly collections showed a similar temporal biting pattern indoors and outdoors (p = 0.912, Fig 4B). There was a substantial probability of sand fly biting in the early evening, which increased during the night, reaching a its maximum around midnight. After that, a drop was observed, with the lowest biting probability just before sunrise. Although Fig 4B shows a clear pattern in the activity, comparison of the biting activity at the different time intervals provided no significant differences between neighboring intervals (S2 Table).

**CDC light trap captures.** A total of 821 sand flies were captured with CDC light traps during seven trapping nights, of which 711 were female and 110 male. The hourly activity pattern of female and male sand flies is depicted in Fig 5. Significantly more female than male sand flies were captured (p < 0.001). The activity of female sand flies between 19h-20h was

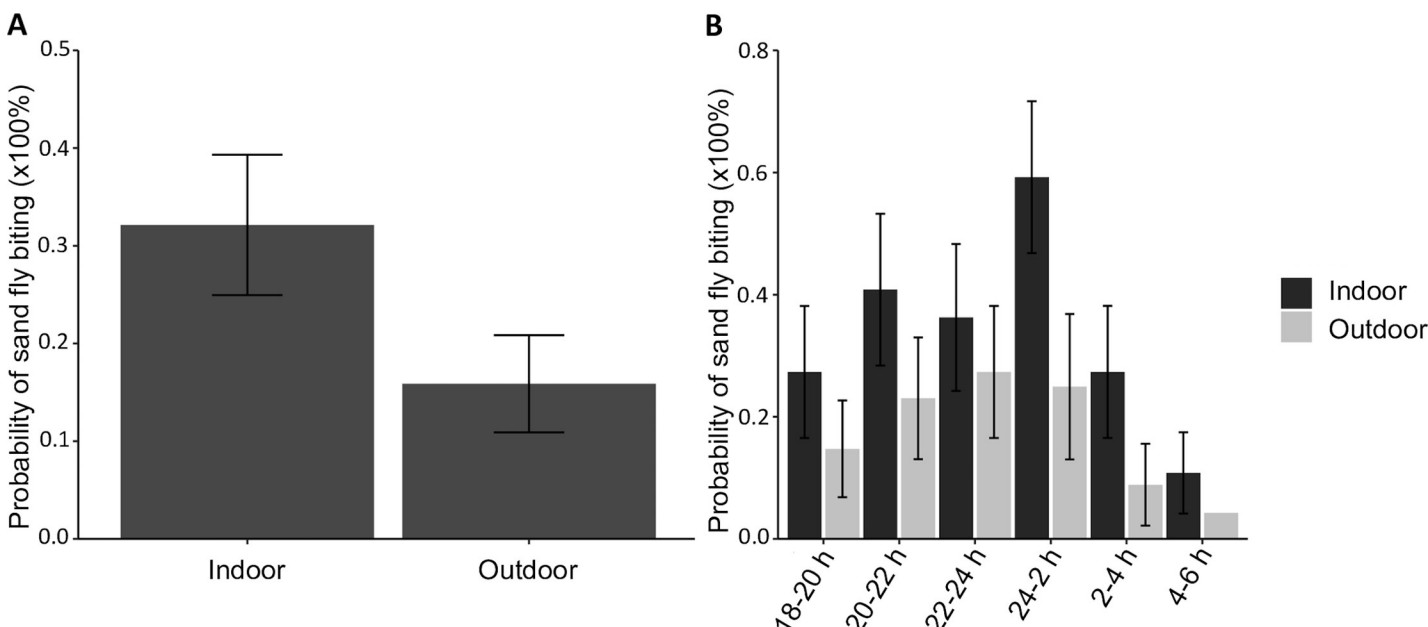

**Fig 4. Indoor and outdoor human biting rhythms of *Phlebotomus pedifer* by human landing catches.** (A) Average probability (%) of sand fly biting indoors and outdoors. (B) Average probability of temporal sand fly biting (%) indoors (dark grey bars) and outdoors (light grey bars). Error bars represent the standard error of the response variable.

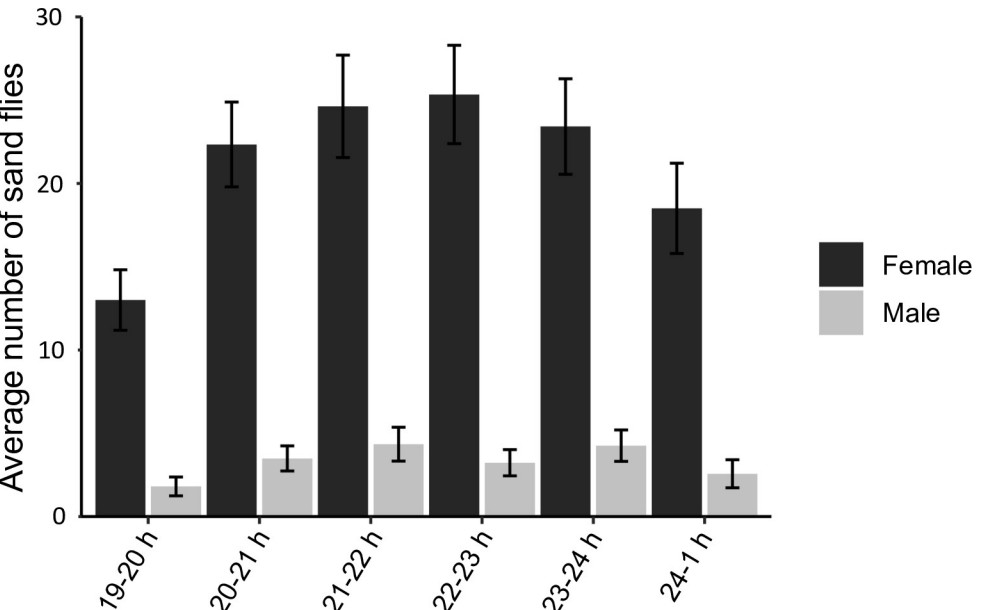

**Fig 5. *Phlebotomus pedifer* activity pattern based on CDC light trap captures.** The left, dark grey and the right, light grey bars are respectively the average number of female and male sand flies at a certain time interval. Error bars represent the standard error of the response variable.

significantly lower compared to the other hours, except for 24h-1h (S3 Table). Other time intervals were not significantly different from each other. Overall, the activity pattern of female sand flies shows that there was considerable activity in the early evening, which increased over time, reaching its maximum at 22h-23h. For male sand flies, no clear trend could be distinguished.

## Host preference

A total of 716 female *P. pedifer* sand flies were used in the host choice experiment, of which in total 65 sand flies were found blood fed over the eight repeats of the experiment (S4 Table).

The *Cyt B* fragment was successfully amplified and sequenced for all freshly engorged sand flies. All sand flies included in the experiment were *P. pedifer*. Fig 6 shows that sand flies were biting both hosts, but significantly more sand flies fed on hyraxes (61.5%) than on humans (38.5%, p = 0.009). The position of the host had no effect on the host choice (p = 0.776).

## Discussion

We gathered novel insights in the biting behavior and activity of *P. pedifer*, which can be used as a guidance in disease control programs; and studied the role of livestock in transmission of CL in southwestern Ethiopia.

We identified the blood meal sources of sand flies in Ochollo indoors, outdoors and in caves. Sand flies acquired their blood meals from hosts of 12 different genera, which is a wider variety compared to the results of Ashford *et al.* (1973) from Ochollo, who found only hyraxes and humans as blood meal sources in caves and indoors [12]. This may be linked to our larger sample size and the availability of more sophisticated analysis methods.

Overall, the majority of sand flies fed on humans and 4.7% of these sand flies were *Leishmania* DNA positive. Additionally, previous research showed that there is a high infection prevalence in humans in Ochollo and a study in Kenya demonstrated that *L. aethiopica*

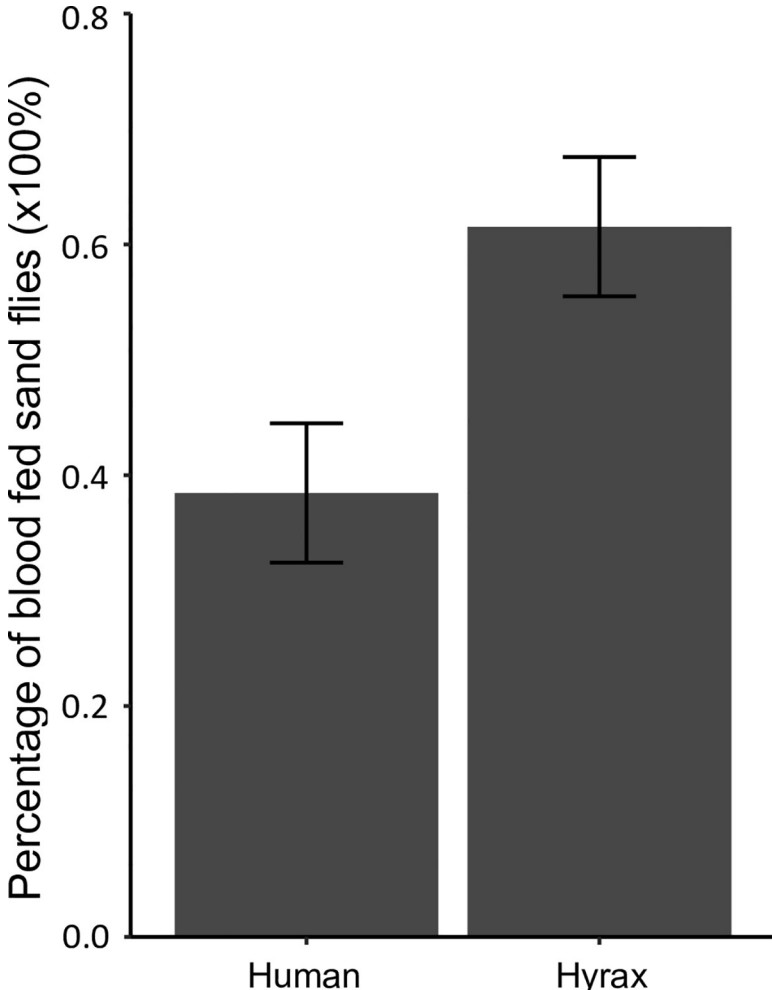

**Fig 6. Host choice preference of *Phlebotomus pedifer* in an experimental set-up.** Average percentage of sand flies that fed on a human or hyrax host during the host choice experiment. Error bars represent the standard error of the response variable.

transmission from a human CL lesion to *P. pedifer* is very efficient [8,14]. These combined data suggest that humans are probably more than just an accidental host in the transmission cycle [8,14,19]. Only 10.6% of the sand fly blood meals were derived from hyraxes, however, the low representation of hyraxes in the blood meals may be biased by the higher proportion of specimens captured indoors in the analysed material. None of the sand flies that fed on hyraxes were found kDNA positive, which is an interesting result because we recently documented that 20% (5/25) of the hyraxes captured in Ochollo were *Leishmania* DNA positive [13]. Although CL in Ethiopia has only been reported as zoonotic with hyraxes serving as the only reservoir host [5,6,19,31], this study suggests that human-to-human transmission may be involved in southwestern Ethiopia. Hence, control should focus on humans, whether or not with additional reservoir control. Notably, the kDNA positive blood fed sand flies should be interpreted with care, because sand flies could have been infected before the current blood meal was acquired.

Some sand flies fed on rodents, in particular on different spiny mouse species (*Acomys spp.*), thicket rat (*Grammomys sp.*) and grass rat (*Arvicanthis sp.*). Several researchers have

focused already on rodents as potential reservoirs of CL in Ethiopia. In a previous study that we carried out in Ochollo (2019), only a single African pigmy mouse (*Mus mahomet*) out of 192 rodents of eight different species was found kDNA positive. Despite the large trapping effort in that study, *Acomys* and *Grammomys spp.* were not captured [13]. In another study carried out all over Ethiopia by Kassahun and his colleagues (2015), 141 *Acomys spp.* were collected, of which 14 (9.9%) were found kDNA positive and three of these could be further identified as *L. tropica* infections [32]. A giant rat (*Cricetomys sp.*) and a ground squirrel (*Xerus rutilus*) have been found naturally infected with *L. aethiopica*. The latter was found in Aba Roba (1200 m), a visceral leishmaniasis (VL) endemic area in Ethiopia, where human CL cases have never been reported [33,34]. Except these observations, *L. aethiopica* has to our knowledge never been found in rodents before, despite the various sampling efforts that have been undertaken previously [5,12,34,35]. This suggests that rodents are probably not a reservoir for *L. aethiopica* and hence do not play an important role in the transmission dynamics.

*Acomys spp.* in Ethiopia are known to inhabit rocky slopes and rock crevices, but in our study, five out of six sand flies that fed on this species were found indoors [36]. Likewise, three sand flies that were captured indoors had fed on hyraxes. This result suggests that sand flies might rest indoors after have taken their blood meal elsewhere, which could be further investigated with i.e. fluorescent powder on sand fly wings to demonstrate their dispersal [37,38].

Remarkably, the blood meal analysis from cave collected sand flies demonstrated that sand flies feed on humans as much as on hyraxes, while hyraxes are abundant and live inside the caves. This could be interpreted as an increased preference for biting humans. We tested this by a host choice experiment, in which human and hyrax were both available. Surprisingly, while sand flies do feed on both hosts, there was a significant preference for hyraxes, which contradicts the previous hypothesis. The result that humans are equally dominant as hyraxes as sand fly host meals in caves is probably not due to blood meal preference, but potentially to an increased availability of humans during the peak sand fly activity hours. Previously, Ashford *et al.* recommended complete hyrax elimination by shooting or biological control, such as release of predators [39]. Other researchers suggested hyrax elimination near human settlements (about 1 km) as a possible intervention against *L. aethiopica* transmission [31,40]. This rises the concern that *P. pedifer*'s preferred blood meal host would not be available anymore, resulting in a shift towards biting humans, thereby increasing their exposure to sand fly bites and accordingly their risk of infection. A study of Svobodova *et al.* (2006) showed that asymptomatically infected hyraxes were infectious to *P. arabicus*, but with a low success rate [41]. Additional research remains necessary to establish the transmission efficiency of parasites from infected hyraxes to the current vector to deliberate whether elimination of hyraxes should be included in control programs. The fact that that sand flies captured from caves obtained a similar proportion of blood meals from humans as from hyraxes implies that humans are accessible as blood source in proximity to the hyrax habitats.

Many specimens in the blood meal analysis did not provide a successful PCR or sequence according to the previously set requirements, while host sequences could be determined from all freshly fed sand flies in the host choice experiment. It has been shown that the success rate of host DNA analysis is negatively correlated with the time-course after the blood meal was taken [15,42–44]. We did not record the estimated days post-feeding, but sand flies with partially digested blood were included in the blood meal analysis, which explains the success rate of the blood meal analysis.

To unravel the complex CL cycle in southwestern Ethiopia, it is important to assess all players of transmission. This study demonstrates that livestock accounts for 21.1% of the blood meal sources of *P. pedifer*, but in ear and nose biopsies from goats and bovines, we found only a single goat nose biopsy positive for kDNA and SL-RNA with a high Ct value. This points to a

relatively low, but viable parasitemia, although persistence and transmission of the parasites are not guaranteed [25]. Overall, it should be considered that some animals in the current study might have had parasites in their skin, which remained undetected due to the collection of only a small tissue biopsy [45].

Studies have already found DNA or antibodies indicating the presence of VL parasites in livestock, also in northern Ethiopia [46–49]. Research investigating the role of livestock in CL transmission is rather scarce. A study conducted during a CL outbreak in a non-endemic village in Venezuela found suspected active CL lesions in seven out of 29 (24%) donkeys in hairless areas (ear, tail, etc.), of which six lesion samples contained *Leishmania* parasites [50]. In a similar research conducted in a CL endemic area in Kenya, one goat was found with lesions and detectable levels of *L. aethiopica* DNA in the skin and other organs [51].

Based on our results, gathered from a large sample size collected from areas with different ecological features and screened with highly sensitive assays, we conclude that domestic animals in similar ecological areas in southwestern Ethiopia are likely not to play a considerable role in transmission. However, many sand flies acquired their blood meal from these animals and it was observed that livestock is living close to or even inside human settlements in Ochollo. It has been suggested to keep livestock close to human settlements to divert vector biting from humans (zooprophylaxis) or to use them as baits for vector attraction to insecticide-treated livestock [52–55]. In contrast, other researchers assert that this could increase the vector population near humans (zoopotentiation) or augment the vector infectivity if blood meal sources are readily available [47,53,55,56]. More research is necessary to determine whether domestic animals could serve as protection against contraction of leishmaniasis.

Understanding the vector's biting behavior gives an indication about when and where *Leishmania* transmission occurs, and at which time and place control strategies would be most effective. Both activity experiments showed that sand flies are predominantly active around midnight and the majority of the sand flies were captured indoors with HLC. Therefore, insecticide-treated bed nets or indoor residual spraying are potentially effective control strategies to manage the peak transmission at night [57–60]. Considerable activity was also observed in the early evening with about 30% of the sand flies captured outdoors by HLC. During the fieldwork, children were collecting water near caves and rock crevices and adults were performing outdoor activities in the early evening (e.g. dinner preparation and washing), thereby increasing their risk of exposure to potentially infectious sand fly bites. This was also shown in a study by Sang *et al.* in a CL endemic area in Kenya, where almost all CL cases admitted that they often visit caves [61]. Hence, improvement of community knowledge and attempts to decrease the vector population densities near places of outdoor activity could contribute to a reduction of residual transmission [62].

The activity of the CL vectors in Ethiopia has never been studied so far, but similar studies were carried out on *P. orientalis* in different VL foci in northern Ethiopia and Sudan [63–68]. These studies found various activity patterns for this vector species, indicating that the activity of a single species can differ between regions. Research on sand fly behavior in each ecologically different setting is accordingly necessary to accomplish efficient vector control.

In conclusion, this study shows that sand flies in Ochollo often feed on humans and, therefore, human-to human transmission of *L. aethiopica* should be considered. Hyraxes are the preferred blood meal source when hosts are equally accessible, so the efficiency of parasite transmission from *H. brucei* to *P. pedifer* should be investigated before including them in control programs. Livestock appears an important blood meal source for sand flies, but does probably not play a significant role in transmission of CL in southwestern Ethiopia. *P. pedifer* is mainly active at night indoors, but there is also considerable outdoor activity, suggesting that combined measures are required for efficient disease control.

## Supporting information

**S1 Table. Comparison of different blood meal sources of Phlebotomus pedifer overall and in each habitat (indoor, outdoor, cave) separately.**
(PDF)

**S2 Table. Comparisons of hourly differences in human biting behavior of *Phlebotomus pedifer* sand flies indoors and outdoors by human landing catches.**
(PDF)

**S3 Table. Comparisons of hourly differences in activity of male and female *Phlebotomus pedifer* sand flies by means of CDC light trap captures.**
(PDF)

**S4 Table. Overview of the sand fly blood meal sources in the host choice experiment.** For each of the eight iterations of the experiment, the number (%) of sand flies that were used for the experiment, that eventually took a blood meal and which host they were found to feed on are presented.
(PDF)

**S1 Data. Dataset of blood meal sources of *Phlebotomus pedifer*, captured in different months and different habitats.**
(XLSX)

**S2 Data. Binomial dataset of sand fly activity indoors and outdoor based on human landing catches (HLC).**
(XLSX)

**S3 Data. Dataset of activity of cave collected *Phlebotomus pedifer* sand flies based on CDC light trap captures.**
(XLSX)

**S4 Data. Dataset of host choice experiment.**
(XLSX)

## Acknowledgments

We are very grateful to the village head, agricultural extension worker, volunteers and field workers of Ochollo, who made the sample collection possible. Also, we want to thank Dr. Simon Shibru of Arba Minch University for his support in Ethiopia. Our special thanks goes out to Natalie Van Houtte of the Evolutionary Ecology Group (University of Antwerp, Belgium) for her excellent technical assistance in the laboratory.

## Author Contributions

**Conceptualization:** Myrthe Pareyn, Abena Kochora, Teklu Wegayehu, Herwig Leirs, Fekadu Massebo.

**Data curation:** Myrthe Pareyn, Abena Kochora, Luca Van Rooy, Nigatu Eligo, Nigatu Girma.

**Formal analysis:** Myrthe Pareyn, Abena Kochora, Luca Van Rooy, Nigatu Eligo.

**Funding acquisition:** Louis Maes, Guy Caljon, Bernt Lindtjørn, Herwig Leirs, Fekadu Massebo.

**Investigation:** Myrthe Pareyn, Abena Kochora, Luca Van Rooy, Nigatu Eligo, Nigatu Girma, Behailu Merdekios.

**Methodology:** Myrthe Pareyn, Abena Kochora, Fekadu Massebo.

**Project administration:** Myrthe Pareyn, Abena Kochora, Luca Van Rooy, Nigatu Girma.

**Resources:** Louis Maes, Guy Caljon, Bernt Lindtjørn, Herwig Leirs, Fekadu Massebo.

**Software:** Myrthe Pareyn, Bram Vanden Broecke.

**Supervision:** Myrthe Pareyn, Abena Kochora, Bram Vanden Broecke, Behailu Merdekios, Teklu Wegayehu, Bernt Lindtjørn, Herwig Leirs, Fekadu Massebo.

**Validation:** Myrthe Pareyn, Bram Vanden Broecke.

**Visualization:** Myrthe Pareyn, Luca Van Rooy, Bram Vanden Broecke.

**Writing – original draft:** Myrthe Pareyn, Abena Kochora.

**Writing – review & editing:** Myrthe Pareyn, Luca Van Rooy, Bram Vanden Broecke, Teklu Wegayehu, Guy Caljon, Bernt Lindtjørn, Herwig Leirs, Fekadu Massebo.

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
