## [Decision Letter · Decision Letter 0]

29 Dec 2019

Dear Dr Pareyn:

Thank you very much for submitting your manuscript "Feeding behavior and activity of Phlebotomus pedifer and potential reservoir hosts of Leishmania aethiopica in southwestern Ethiopia." (#PNTD-D-19-01953) for review by PLOS Neglected Tropical Diseases. Your manuscript was fully evaluated at the editorial level and by independent peer reviewers. The reviewers appreciated the attention to an important problem, but raised some substantial concerns about the manuscript as it currently stands. These issues must be addressed before we would be willing to consider a revised version of your study. We cannot, of course, promise publication at that time.

We therefore ask you to modify the manuscript according to the review recommendations before we can consider your manuscript for acceptance. Your revisions should address the specific points made by each reviewer. 

When you are ready to resubmit, please be prepared to upload the following:

(1) A letter containing a detailed list of your responses to the review comments and a description of the changes you have made in the manuscript.

(2) Two versions of the manuscript: one with either highlights or tracked changes denoting where the text has been changed (uploaded as a "Revised Article with Changes Highlighted" file); the other a clean version (uploaded as the article file).

(3) If available, a striking still image (a new image if one is available or an existing one from within your manuscript). If your manuscript is accepted for publication, this image may be featured on our website. Images should ideally be high resolution, eye-catching, single panel images; where one is available, please use 'add file' at the time of resubmission and select 'striking image' as the file type. 

Please provide a short caption, including credits, uploaded as a separate "Other" file. If your image is from someone other than yourself, please ensure that the artist has read and agreed to the terms and conditions of the Creative Commons Attribution License at http://journals.plos.org/plosntds/s/content-license (NOTE: we cannot publish copyrighted images). 

(4) If applicable, we encourage you to add a list of accession numbers/ID numbers for genes and proteins mentioned in the text (these should be listed as a paragraph at the end of the manuscript). You can supply accession numbers for any database, so long as the database is publicly accessible and stable. Examples include LocusLink and SwissProt.

(5) To enhance the reproducibility of your results, we recommend that you deposit your laboratory protocols in protocols.io, where a protocol can be assigned its own identifier (DOI) such that it can be cited independently in the future. For instructions see http://journals.plos.org/plosntds/s/submission-guidelines#loc-methods

While revising your submission, please upload your figure files to the Preflight Analysis and Conversion Engine (PACE) digital diagnostic tool, https://pacev2.apexcovantage.com/ PACE helps ensure that figures meet PLOS requirements. To use PACE, you must first register as a user. Then, login and navigate to the UPLOAD tab, where you will find detailed instructions on how to use the tool. If you encounter any issues or have any questions when using PACE, please email us at figures@plos.org.

We hope to receive your revised manuscript by Feb 27 2020 11:59PM. If you anticipate any delay in its return, we ask that you let us know the expected resubmission date by replying to this email.

To submit a revision, go to https://www.editorialmanager.com/pntd/ and log in as an Author. You will see a menu item call Submission Needing Revision. You will find your submission record there. 

Sincerely,

Luigi Gradoni

Guest Editor

Jesus Valenzuela

Deputy Editor

Reviewer's Responses to Questions

**Key Review Criteria Required for Acceptance?**

**Methods**

-Are the objectives of the study clearly articulated with a clear testable hypothesis stated?

-Is the study design appropriate to address the stated objectives?

-Is the population clearly described and appropriate for the hypothesis being tested?

-Is the sample size sufficient to ensure adequate power to address the hypothesis being tested?

-Were correct statistical analysis used to support conclusions?

-Are there concerns about ethical or regulatory requirements being met?

Reviewer #1: Some experiments need more detailed description, see my Summary and general comments.

Reviewer #2: The study is well designed and methods fully and clearly described. The only point at issue is the fact that sand fly collection for bloodmeal analysis was not performed equally in three different habitats: CDC light traps and sticky traps were used indoors and in caves while outdoors, only sticky traps were used. In addition, collections in caves were performed in different months than collections indoors and outdoors. This heterogeneity is understandable but authors must interpret the results appropriately (more below).

Reviewer #3: The objectives of the study are clearly articulated with the clear hypothesis. The study design is appropriate to the objectives, even if some additional procedure should be applied (eg Leishmania sp identiifcation by PCR-RFLP analysis). Population and sample size are appropriate and adequate for the hypothesis tested. Correct statistical analysis support the conclusions and ethical requirements are being met.

**Results**

-Does the analysis presented match the analysis plan?

-Are the results clearly and completely presented?

-Are the figures (Tables, Images) of sufficient quality for clarity?

Reviewer #1: Results are clearly and completely presented but I am missing data about temperature and relative humidity during the experiments on sand fly activity, for more details see Summary and general comments.

Reviewer #2: Results are completely presented and all the figures have sufficient quality. However, the summary evaluation of blood meal sources (lines 289-294) is biased by higher representation of the indoor sample than both outdoor and cave samples. I suggest including the information that 129 from180 sand files came from indoor captures to the legend of the Table 1.

Reviewer #3: The analysis matches to the analysis plan. Due to the fact that cutaneous leishmaniasis (CL) in Ethiopia are caused prevalently by L. aethiopica but also by L. tropica and L.major it is important that the AA identify Leishmania at species level and not simply to genus level, even simply by PCR-RFLP procedure. Otherwise the Leishmania molecular analysis remains incomplete. The results are clearly presented as the quality and clarity of Figures and Tables.

**Conclusions**

-Are the conclusions supported by the data presented?

-Are the limitations of analysis clearly described?

-Do the authors discuss how these data can be helpful to advance our understanding of the topic under study?

-Is public health relevance addressed?

Reviewer #1: Conclusions are the weakest part of the manuscript but could be easily corrected, see Summary and general comments.

Reviewer #2: I do not agree with the conclusion that Leishmania transmission in SW Ethiopia is likely mainly anthroponotic (see the general comments and comments to discussion).

Reviewer #3: All the aspects of the conclusions are clearly supported and discussed, included the limitations of the analysis. The authors describe well the helpful to advance the topic under study. It is clear a paper of public health relevence in terms of surveillance and control of cutaneous leishmaniasis in Ethiopia.

**Editorial and Data Presentation Modifications?**

Reviewer #1: Abstract 

line 23-24: authors did not establish but studied the role of livestock in CL transmission.

Line 32: indicate if all 180 were P. pedifer females.

Line 38: “amount” . Do you mean “number” or “proportion”?

Lines 41-42: please correct the conclusions: 1) use more precise explanation about proposed transmission cycle (e.g. ..very likely, also anthroponotic transmission occurs in Ochollo”), see also my general comment. 2) Livestock was proposed as a potential reservoir for VL but never for CL, therefore the sentence needs to be corrected also in this aspect.

Background

line 73-74: Mutinga and Odhiambo demonstrated susceptibility of P. pedifer to L. aethiopica. They did not study vectorial capacity. Moreover, susceptibility only may imply/suggest the efficient vector. Please, change the sentence accordingly. 

Materials and Methods

Line 125-6: location of caves should be specified more. Are they really in the village? How far is from the caves to the nearest houses?

Line 221: explain better the position of the human volunteer during the experiment. How the human volunteer was separated from the opposite cage with the hyrax? 

In the revised version please correct the format, in some pages (e.g. lines 137, 159 and elsewhere) the subtitles were merged with the text (please place the text on the new line).

Results

Line 286: I suggest correction “..sand fly species of subgenus Larroussius” 

Discussion

Line 467: I suggest introducing somehow the paragraph, e.g. by “In conclusion….”.

References

Please check names of the authors again, you give three references by Ashford but only one of them is correct (Ashford RW, not Ashford W, as written in refs. 12 and 38).

Reviewer #2: Abstract - principal findings

Lines 32-35: Second and third sentences should be changed to: Humans were the predominant blood meal source indoors (65.9%, P <0.001) while no significant differences in blood meal sources were determined outdoors and in caves. In caves, hyraxes were represented in blood meals equally as humans (45.5% and 42.4 %, respectively), but the host choice experiment revealed….. 

Abstract - conclusions

Line 42: please change the conclusion that Leishmania transmission in SW Ethiopia is likely mainly anthroponotic (see the general comments and comments to discussion). 

Author summary

Lines 51-52 the sentence “humans are likely the main source of the infection “should be changed to “humans are an important source of the infection”

Methods

Line 137 Blood isolation from the blood fed specimens („from“ is missing)

Line 215 – citation of the original description of the method is missing (I suppose Sadlova et al. 2003, https://doi.org/10.1046/j.1365-2915.2003.00434.x.)

Discussion

Lines 370-371. Sentence „ On the contrary, only 10.6% of the sand fly blood meals were derived from hyraxes and none of these sand flies were found kDNA positive should be followed by the sentence „However, the low representation of hyraxes in the blood meals may be biased by high prevalence of specimens captured indoor in the analysed material.“ and also the following sentences adapted according to this fact.

Lines 374-375 Instead of ”..this study demonstrates that there is very likely also anthroponotic transmission..” should be written “ ..this study suggests that human-to-human transmission may be involved ..”

Line 467 I suggest to write „often feed on humans and, therefore, human-to-human transmission should be considered” instead of „mainly feed on humans and that there is likely also anthroponotic transmission”

Reviewer #3: The paper of Pareyn et al 2019 (PLOS NTD) appears to play the role of prerequisite of the present survey and not simply as a reference.

**Summary and General Comments**

Reviewer #1: Authors studied activity and feeding preferences of P. pedifer in the well-known Ethiopian focus of cutaneous leishmaniasis caused by L. aethiopica. In addition, they search for Leishmania presence in livestock by molecular methods. Results are interesting and important; however, their interpretation needs some corrections and some more detailed description of the experiments would be useful:

1. I appreciate that the work was done in natural conditions but it would be useful to describe some experiments in more details, see my specific comments. More importantly, I am missing data about ambient temperature and relative humidity during the study on biting patterns and CDC light trap captures (Lines 331-340). Sand fly activity is highly dependent on these parameters and I expect that some differences observed might be explained by differences in humidity and temperature. Did you use data loggers and could you add these data? 

2. The study shows that livestock has no role in CL transmission but serve as a blood source. This is just a confirmation of other previous studies on CL. Therefore, in the Abstract two sentences in Background (lines 23-24) and conclusion (lines 41-42) should be corrected. Same is required for the sentence in the Background on lines 88-89. Bovines are frequently bitten by sand flies in many countries, e.g. in East Mediterranean or Magreb, but they do not serve as reservoirs there. Please, make all these corrections. 

3. Similarly, in Discussion (lines 358-9) there is the sentence “We gathered novel insight in …..the role of livestock in transmission of CL in southwestern Ethiopia, which can be used as a guidance in disease control programs”. What is the novel insight concerning the livestock? I did not find any. What do you mean by this general sentence? Does it mean that the livestock should be sprayed or not? 

4. Authors bring the evidence that many P. pedifer feed on humans, which are probably the most common and most easily accessible mammals in the area. This, however, does not mean that the disease is anthroponotic, it seems to me that there is a combination of anthroponotic and zoonotic transmission, as correctly mentioned by authors in Discussion (lines 374-5: “… that there is very likely also anthroponotic transmission….). Please, change the conclusion on lines 41-42 accordingly.

Reviewer #2: The study is aimed at feeding preferences and biting activity of sand flies (Phlebotomus pedifer) in the CL foci in SW Ethiopia caused by L. aethiopica. In addition, Leishmania presence in livestock (goats and bovines) was studied. The method used is appropriate and the results are novel and interesting. Valuable is the description of the broad host spectrum of P. pedifer and detailed description of the temporal biting pattern of this sand fly species. However, the summary interpretation of the data and some conclusions need revision. Authors speculate about the anthroponotic transmission of L. aethiopica in Ochollo based on the fact that majority (59%) of sand flies fed on humans and only 10% on hyraxes. However, in their samples, 129 flies were captured indoors in comparison with only 18 captured outdoor and 33 in caves. Importantly, representation of human blood was not significantly higher in outdoor and cave samples and in direct preference test, P. pedifer preferred hyraxes over humans. My specific suggestions are written in comments to respective parts of the manuscript.

Reviewer #3: The manuscript on the “Feeding behavior and activity of Phlebotomus pedifer and potential reservoir hosts of Leishmania aethiopica in southwestern Ethiopia” is interesting and well structured. However, the survey, based on molecular approaches, suffers sometime of the no application of parasitological classical procedures. In addition the paper of Pareyn et al 2019 appears to play the role of prerequisite of the present survey and not simply as a reference. For example, rodents analysis was better evaluated in Pareyn et al as the Leishmania species identification.

Due to the fact that cutaneous leishmaniasis (CL) in Ethiopia are caused prevalently by L. aethiopica but also by L. tropica and L. major it is important that the AA identify Leishmania at species level and not simply to genus, even simply by PCR-RFLP procedure. Otherwise the Leishmania molecular analysis remains incomplete. Hyrax xenodiagnosis studies should be implemented to solve the L. aethiopica role of vector and reservoir hosts.

Since the Results the text appears not accurate, as the rest of the manuscript, and sometimes it is confused. 

Specific comments

Lines 23: AA should insert “Southern Ethiopia”, the geographic location of the area studied.

Line 46: Correct “is caused” in “is mainly caused”. Three Leishmania species (without consider L. donovani….) cause CL in Ethiopia.

Line 50 : What intend the AA as “other potential hosts”? Reservoir Hosts? Please clarify.

Line 72-81: Paper “13” is basic for the project planning. This should be a prerequisite in the Introduction.

Lines 139: The AA should specify the Leishmania sp infected mice. If it is L. major, as in Pareyn et al 2019, give species and ref, otherwise Leishmania species should be given complete of the strain WHO code.

Line 181-182: These 2 lines could be removed doing follow the appropriate reference from line 172 (ref 25 for RT-qPCR SL-RNA, ref 13 for qPCR kDNA).

Line 279: The AA should give the total of the collected sand flies from the 72 households. Total number could justify because sand flies were not identified by morphological keys.

Line 293: The sand flies that fed on humans are 107 not 137. Please correct.

Line 369-370: Ref 14 concerns L. aethiopica but in Kenya.

Lines 371-73: The comment is due by the structure of the work. In the present manuscript human role was one of the principal objectives, different structure was in Pareyn et al 2019. 

Line 382: The AA should report ref 13 following “(2019)”.

PLOS authors have the option to publish the peer review history of their article (what does this mean?). If published, this will include your full peer review and any attached files.

Reviewer #1: No

Reviewer #2: No

Reviewer #3: No

---

## [Decision Letter · Decision Letter 1]

19 Feb 2020

Dear Myrthe,

Thank you very much for submitting your manuscript "Feeding behavior and activity of Phlebotomus pedifer and potential reservoir hosts of Leishmania aethiopica in southwestern Ethiopia." for consideration at PLOS Neglected Tropical Diseases. As with all papers reviewed by the journal, your manuscript was reviewed by members of the editorial board and by several independent reviewers. The reviewers appreciated the attention to an important topic. Based on the reviews, we are likely to accept this manuscript for publication, providing that you modify the manuscript according to the review recommendations. 

Please modify the text as required by the Reviewer 1's comment, with which I agree, copied below:

Authors significantly improved the manuscript and I agree with all but one correction they did. The term “anthroponosis” should be use only if humans are the sole reservoir hosts. Therefore, I think that at least the sentence in the abstract: “In contrast to earlier suggestions of exclusive zoonotic Leishmania transmission, we indicate that CL transmission is very likely also anthroponotic in southwestern Ethiopia.” should be changed to e.g. “In contrast to earlier suggestions of exclusive zoonotic Leishmania transmission, we propose that there is also human-human transmission of CL in southwestern Ethiopia.”

Sincerely,

Luigi

Luigi Gradoni

Guest Editor

Jesus Valenzuela

Deputy Editor

Please modify the text as required by the Reviewer 1's comment, with which I agree, copied below:

Authors significantly improved the manuscript and I agree with all but one correction they did. The term “anthroponosis” should be use only if humans are the sole reservoir hosts. Therefore, I think that at least the sentence in the abstract: “In contrast to earlier suggestions of exclusive zoonotic Leishmania transmission, we indicate that CL transmission is very likely also anthroponotic in southwestern Ethiopia.” should be changed to e.g. “In contrast to earlier suggestions of exclusive zoonotic Leishmania transmission, we propose that there is also human-human transmission of CL in southwestern Ethiopia.”

Reviewer's Responses to Questions

**Key Review Criteria Required for Acceptance?**

**Methods**

-Are the objectives of the study clearly articulated with a clear testable hypothesis stated?

-Is the study design appropriate to address the stated objectives?

-Is the population clearly described and appropriate for the hypothesis being tested?

-Is the sample size sufficient to ensure adequate power to address the hypothesis being tested?

-Were correct statistical analysis used to support conclusions?

-Are there concerns about ethical or regulatory requirements being met?

Reviewer #1: (No Response)

Reviewer #2: (No Response)

Reviewer #3: (No Response)

**Results**

-Does the analysis presented match the analysis plan?

-Are the results clearly and completely presented?

-Are the figures (Tables, Images) of sufficient quality for clarity?

Reviewer #1: (No Response)

Reviewer #2: (No Response)

Reviewer #3: (No Response)

**Conclusions**

-Are the conclusions supported by the data presented?

-Are the limitations of analysis clearly described?

-Do the authors discuss how these data can be helpful to advance our understanding of the topic under study?

-Is public health relevance addressed?

Reviewer #1: (No Response)

Reviewer #2: (No Response)

Reviewer #3: (No Response)

**Editorial and Data Presentation Modifications?**

Reviewer #1: (No Response)

Reviewer #2: (No Response)

Reviewer #3: (No Response)

**Summary and General Comments**

Reviewer #1: Authors significantly improved the manuscript and I agree with all but one correction they did. The term “anthroponosis” should be use only if humans are the sole reservoir hosts. Therefore, I think that at least the sentence in the abstract: “In contrast to earlier suggestions of exclusive zoonotic Leishmania transmission, we indicate that CL transmission is very likely also anthroponotic in southwestern Ethiopia.” should be changed to e.g. “In contrast to earlier suggestions of exclusive zoonotic Leishmania transmission, we propose that there is also human-human transmission of CL in southwestern Ethiopia.”

Reviewer #2: I’m fully satisfied with the revision of data interpretation and recommend the manuscript to publication in PNTD.

Reviewer #3: (No Response)

PLOS authors have the option to publish the peer review history of their article (what does this mean?). If published, this will include your full peer review and any attached files.

Reviewer #1: No

Reviewer #2: No

Reviewer #3: No
---

## [Editor Report · Decision Letter 2]

25 Feb 2020

Dear Myrthe

We are pleased to inform you that your manuscript 'Feeding behavior and activity of Phlebotomus pedifer and potential reservoir hosts of Leishmania aethiopica in southwestern Ethiopia.' has been provisionally accepted for publication in PLOS Neglected Tropical Diseases.

Before your manuscript can be formally accepted you will need to complete some formatting changes, which you will receive in a follow up email. A member of our team will be in touch within two working days with a set of requests.

Best regards,

Luigi

Luigi Gradoni

Guest Editor

Jesus Valenzuela

Deputy Editor

None

---

## [Editor Report · Acceptance letter]

11 Mar 2020

Dear Drs. Pareyn,

We are delighted to inform you that your manuscript, "Feeding behavior and activity of Phlebotomus pedifer and potential reservoir hosts of Leishmania aethiopica in southwestern Ethiopia.," has been formally accepted for publication in PLOS Neglected Tropical Diseases.

Best regards,

Serap Aksoy

Editor-in-Chief

Shaden Kamhawi

Editor-in-Chief
